# High-throughput viable circulating tumor cell isolation using tapered-slit membrane filter-based chipsets in the differential diagnosis of ovarian tumors

**Nam Kyeong Kim**[1,2¤]**, Dong Hoon Suh**[1,2]**, Kidong Kim**[1,2]**, Jae Hong No**[1,2]**, Yong Beom Kim**[1,2]***, Minki Kim**[3]**, Young-Ho Cho**[3]

1 Department of Obstetrics and Gynecology, Seoul National University College of Medicine, Seoul, Korea, 2 Department of Obstetrics and Gynecology, Seoul National University Bundang Hospital, Seongnam, Korea, 3 Cell Bench Research Center, Department of Bio and Brain Engineering, Korea Advanced Institute of Science and Technology (KAIST), Daejeon, Korea

¤ Current address: Department of Obstetrics and Gynecology, Korea University Ansan Hospital, Ansan, Korea
* ybkimlh@snubh.org

**Data Availability Statement:** Data contain potentially identifying or sensitive patient information and cannot be shared publicly because

## Abstract

### Objective

To evaluate the diagnostic performance of circulating tumor cells (CTCs) using tapered-slit membrane filter (TSF)-based chipsets for the differential diagnosis of adnexal tumors.

### Methods

A total of 230 women with indeterminate adnexal tumors were prospectively enrolled. The sensitivity, specificity, and accuracy of the CTC-detecting chipsets were analyzed according to postoperative pathological results and compared with those of cancer antigen (CA)-125 and imaging tests.

### Results

Eighty-one (40.3%) benign tumors, 31 (15.4%) borderline tumors, and 89 (44.3%) ovarian cancers were pathologically confirmed. The sensitivity, specificity, and accuracy of CTC-detecting chipsets (75.3%, 58.0%, and 67.1%) for differentiating ovarian cancer from benign tumors were similar to CA-125 (78.7%, 53.1%, and 66.5%), but lower than CT/MRI (94.2%, 77.9%, and 86.5%). "CTC or CA125" showed increased sensitivity (91.0%) and "CTC and CA-125" revealed increased specificity (77.8%), comparable to CT/MRI. CTC detection rates in stage I/II and stage III/IV ovarian cancers were 69.6% and 81.4%, respectively. The sensitivity to detect high-grade serous (HGS) cancer from benign tumors (84.6%) was higher than that to detect non-HGS cancers (68.0%).

of privacy law. Data are available from the Seoul National University Bundang Hospital Institutional Data Access/Ethics Committee (contact via drb@snubh.org) for researchers who meet the criteria for access to confidential data.

**Funding:** This work was supported by the national R&D project funded by the Ministry of Trade, Industry, and Energy of Korea (Project number 10078295). The funders had no role in the study design, data collection and analysis, decision to publish, or preparation of the manuscript.

**Competing interests:** The authors have declared that no competing interests exist.

## Conclusion

Although the diagnostic performance of the TSF platform to differentiate between ovarian cancer and benign tumors did not yield significant results, the combination of CTC and CA-125 showed promising potential in the diagnostic accuracy of ovarian cancer.

## Introduction

When an adnexal mass is found, surgical treatment direction may vary depending on the final pathological result; therefore, it is important to differentiate malignant from benign or border-line. Intraoperative rupture of ovarian cancer confined to the unilateral ovary (i.e., stage IA) is associated with poor prognosis in the early stages, upstaging to IC1, and requiring adjuvant chemotherapy [1]. If an adnexal tumor initially presumed to be benign is confirmed to be malignant after ovarian cystectomy or salpingo-oophorectomy, an additional staging surgery should be performed [2]. Approximately 70% of ovarian cancer is diagnosed at an advanced stage, and advanced ovarian cancer has poor prognosis, with a 5-year survival rate of <50% [3, 4]. Therefore, a diagnostic tool capable of the early detection of ovarian malignancies before pathological confirmation through surgery is needed.

Several diagnostic tools can aid in the initial diagnosis of adnexal masses before surgery. Cancer antigen (CA)-125 has been used as a leading protein biomarker for the screening, treatment, and follow-up of ovarian cancer for almost four decades. Elevated serum CA-125 occurs in >90% of advanced ovarian cancer cases and in 50–60% of stage I ovarian cancer cases [5]. CA-125 may also be elevated in benign gynecologic conditions, such as adenomyosis, endometriosis, myomas, and non-gynecologic conditions [6]. Consequently, the high false-positive rate and modest sensitivity of CA-125 contribute to unnecessary surgical procedures and psychological consequences in these women [7]. Imaging tests included ultrasound with Doppler, computerized tomography (CT), and magnetic resonance imaging (MRI) [8]. In addition, methods for predicting the likelihood of malignancy in an adnexal mass, including the Risk of Malignancy Index (RMI) and Risk of Malignancy Algorithm (ROMA), have been reported [9, 10]. However, the routine use of these tools is limited by their inadequate diagnostic performance in certain situations and high costs.

Circulating tumor cells (CTCs) are cancer cells shed from primary or metastatic tumors into the bloodstream. CTC detection has several advantages over tissue biopsy:1) convenience, 2) minimal invasiveness, 3) serial evaluation, and 4) evaluation of the entire tumor burden instead of a limited part [11]. Previous studies have shown that CTCs serve as prognostic factors for overall survival and progression-free survival (PFS) in breast, colorectal, prostate, and lung cancers [12–15]. Several technologies have been developed to isolate CTCs from whole blood using the differential expression of biological factors (tumor-associated antigens or simple markers of epithelial vs. mesenchymal/hematopoietic derivation) or physical properties of cancer cells (size, weight, or density) [11, 16]. CellSearch® system, using a magnetic ferrofluid containing antibodies against epithelial cell adhesion molecule, has been approved by the Food and Drug Administration as a CTC enumeration method in patients with metastatic breast, prostate, and colorectal cancer [17]. However, only epithelial cell types of CTCs can be selected because chemical pre-/post-treatment does not allow other cells to remain viable [18]. We developed a CTC isolation chipset using a tapered-slit membrane filter (TSF) based on multiple physical properties, such as size and elasticity that can detect cell types other than epithelial cells while maintaining CTC viability. Recently, studies have been conducted on CTCs as a

predictive or prognostic factor for ovarian cancer. This study aimed to evaluate CTC isolation using TSF-based chipsets for the differential diagnosis of adnexal tumors.

## Materials and methods

### Study population

A total of 230 patients scheduled to undergo elective surgery for an indeterminate ovarian mass were prospectively recruited at Seoul National University Bundang Hospital from May 27, 2015 to April 4, 2016 and from November 05, 2020 to May 31, 2023. The exclusion criteria were as follows:1) patients with a history of other malignancies within the past five years from enrollment, 2) patients who received neoadjuvant chemotherapy before surgery, 3) patients with recurrent ovarian cancer, and 4) patients of non-ovarian origin in the final pathology report. Finally, 201 patients were analyzed in this study. Written informed consent was obtained from all participants. This study was approved by the institutional review boards of our institution (B-1408-263-003 and E-2008-630-001) and conducted in accordance with the Declaration of Helsinki.

### Blood collection and preparation

After the patient was placed under general anesthesia, 5 mL of peripheral blood was drawn from the antecubital vein before the start of planned surgery. All blood samples were collected in BD Vacutainer tubes and transferred to the Korea Advanced Institute of Science and Technology (KAIST) to identify and count the CTCs. The collected tubes were packed in ice and delivered within 6 hours of sampling to prevent cell lysis and destruction during delivery.

### Data collection

Information on age, final pathological results (benign, borderline, or malignant), ovarian tumor size, International Federation of Gynecology and Obstetrics (FIGO) stage, cancer histology, preoperative CA-125 levels, and presumptive diagnosis on CT or MRI were collected. The 2014 FIGO staging system was used to stage ovarian cancer. The cutoff value for normal CA-125 levels was 35 U/mL. The presence of ascites was defined as moderate-to-severe ascites on preoperative CT or MRI. CTC positivity was defined as the presence of one or more detected CTCs, whereas CTC negativity was defined as the absence of detected CTCs.

### Identification and counting of CTCs

CTC isolation and counting were performed using a tapered-slit filter (TSF) platform [19, 20] (Fig 1). The TSF platform isolates CTCs based on their physical properties, such as size and deformability, irrespective of their surface protein expression. Moreover, owing to its unique slit design (Fig 1) featuring a wide cell entrance and gradually narrowing cell exits, a high flow rate can be achieved while minimizing cell stress. Five mL of blood from an ovarian cancer patient was diluted in 10.0 mL of phosphate-buffered saline (PBS). The diluted sample was then directly processed into the TSF platform using a syringe pump for withdrawal without undergoing any pretreatment, such as Ficoll separation or cell fixation. After filtration, the captured cells were carefully released by applying a reverse flow of PBS using a syringe pump. The released cells were mounted onto glass slides using a cytocentrifuge (Shandon Cytospin III; Thermo Scientific, Wilmington, DE, USA). Cell-mounted glass slides were fixed, permeabilized, blocked, and subjected to immunofluorescent staining [20]. Subsequently, the fluorescent images (Fig 2) were acquired using a fluorescence microscope system (Eclipse Ti, Nikon) and quantified using MetaMorph software (Molecular Devices, Sunnyvale, CA, USA). All

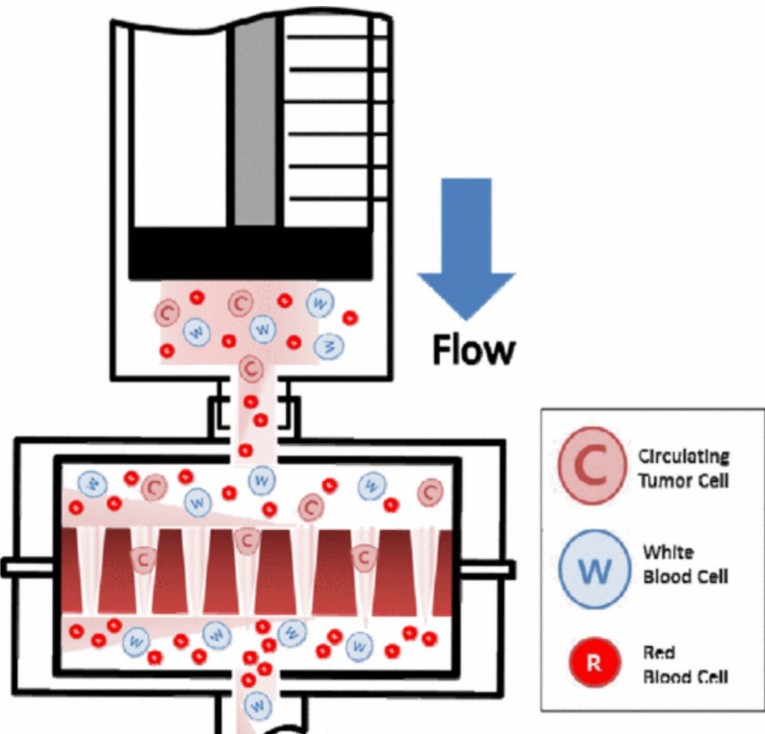

**Fig 1. Design of tapered-slit filter platform.**

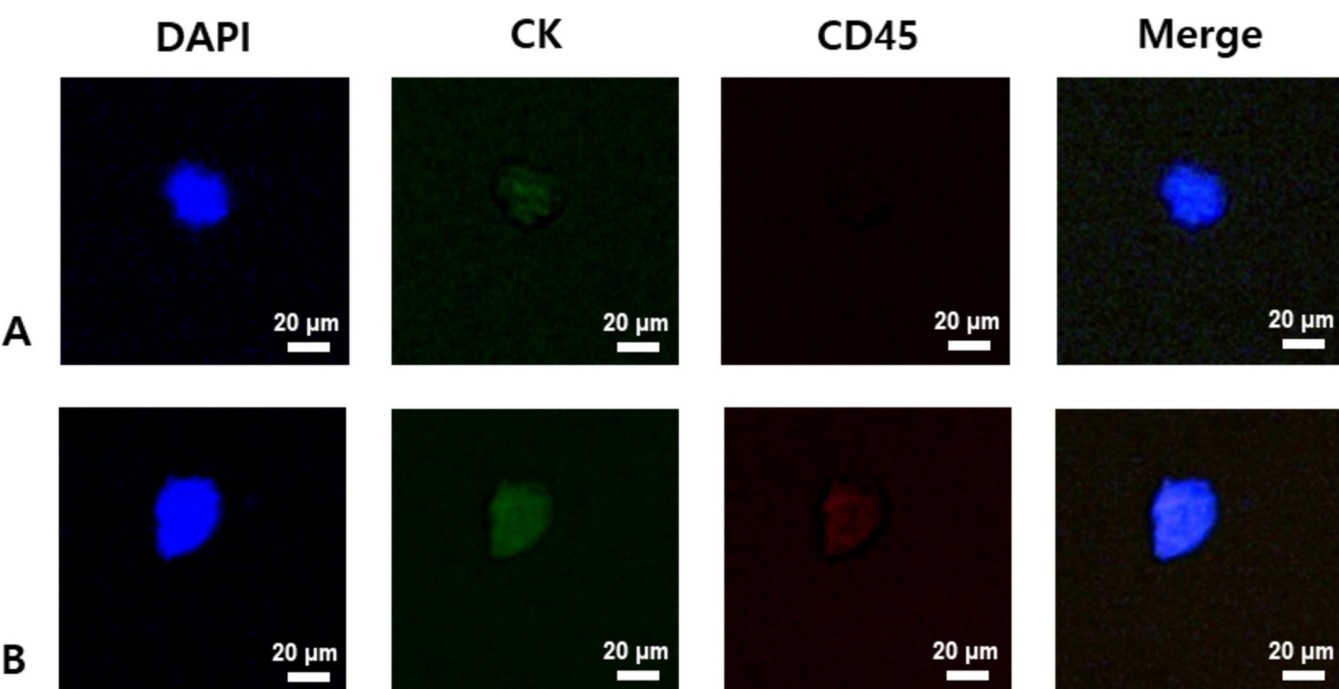

**Fig 2.** Immunofluorescent cell images: A. circulating tumor cell (DAPI+/CK+/CD45-) and B. white blood cell (B, DAPI+/CK±/CD45+) from ovarian cancer patient's blood, where the bar represents 20 μm, DAPI denotes 4′,6-diamidino-2-phenylindole, CD45 indicates the cluster of differentiation 45, and CK means cytokeratin.

immunofluorescent cells (Fig 2) were examined and enumerated meticulously as CTCs based on the criteria of immunostaining (4′,6-diamidino-2-phenylindole, DAPI, positive; cluster of differentiation 45, CD45, negative; cytokeratin, CK, positive), morphology (such as higher degree of irregularity compared to background blood cells), cell size, and nucleus-to-cytoplasm ratio. CTCs were identified and counted from blood samples without patient information.

## Statistical analysis

The sensitivity, specificity, and accuracy of the CTC detection chipsets were analyzed according to postoperative pathologic results and compared with those of other screening tools, such as CA-125 and CT/MRI. The area under the curve (AUC) was calculated to evaluate the ability of CTCs as a screening tool by creating a receiver operating characteristic (ROC) curve. Pearson's $\chi^2$ test or Fisher's exact test was used to compare categorical variables, while Student's t-test or Mann–Whitney test was used to compare continuous variables. Statistical analyses were performed using IBM SPSS Statistics for Windows (version 27.0; IBM Corp., Armonk, NY, USA) and MedCalc (version 22). Statistical significance was set at a P-value < 0.05.

## Results

A total of 201 patients were analyzed in this study. The clinical characteristics of the study population are shown in Table 1, and patient characteristics according to the final pathological reports after surgery are shown in Table 2. Eighty-one (40.3%), 31 (15.4%), and 89 (44.3%) were pathologically confirmed with benign, borderline, and malignant tumors, respectively. The mean age of the study population was 48.0 years and the mean tumor size was 11.4 cm. The frequency of age > 48 years (60.7% vs. 34.6%, p = 0.001) and the frequency of tumor size > 11cm (46.1% vs. 24.7%, p = 0.004) were significantly higher in the malignant group compared to the benign group. The frequency of abnormal CA-125 was significantly higher in the malignant group compared to the benign and borderline groups (malignant vs. benign, 78.7% vs. 46.9%, p<0.001; malignant vs. borderline, 78.7% vs. 22.6%, p<0.001). Of 89 patients with malignant ovarian tumors, 46 (51.7%) were diagnosed at early stage (I and II) and 43 (48.3%) were diagnosed at advanced stage (III and IV). The most common histologic type was high-grade serous (HGS) carcinoma (43.8%), followed by clear cell carcinoma (19.1%) and endometrioid carcinoma (16.9%). In the comparison between the benign and stage I/II ovarian cancer groups, the significant difference in the frequency of abnormal CA -125 disappeared (S1 Table). There were significant differences in the frequencies of age > 48 years, tumor size > 11 cm, and malignant diagnoses on CT/MRI.

In the study population, the median number of CTCs was 1.0 (range, 0–23) and CTCs were detected in 116 patients (57.7%). Table 3 shows a comparison of preoperative CTCs according to the final pathological results. The mean number of CTCs in the benign group had no significant differences from those in the overall, early-stage, and advanced-stage cancer groups (mean ± standard deviation, 1.09 ± 2.74 vs. 1.51 ± 1.48 vs. 1.30 ± 1.13 vs. 1.72 ± 1.76). The CTC detection rates in the benign, borderline, and cancer (early stage/advanced stage) groups were 42.0%, 48.4%, and 75.3% (69.6%/81.4%), respectively. These values were significantly different between the benign and cancer groups, regardless of stage, and between the borderline and cancer groups.

Table 4 and Fig 3 show the diagnostic performance of preoperative CTC, CA-125, and CT/MRI for the differential diagnosis of adnexal masses. Preoperative CTC had a sensitivity of 75.3%, specificity of 58.0%, and accuracy of 67.1% for differentiating ovarian cancer from benign ovarian tumors, excluding borderline ovarian tumors. The values were similar to CA-

**Table 1. Baseline characteristics of study population (N = 201).**

| Variables | Values |
| --- | --- |
| Age (years) | 48.0 ± 12.1 |
| CA-125 (U/mL) | 529.3 ± 1702.5 |
| ≤ 35 | 86 (42.8) |
| >35 | 115 (57.2) |
| CT or MRI | |
| Benign | 56 (27.9) |
| Borderline | 26 (12.9) |
| Malignancy | 112 (55.7) |
| Not done | 7 (3.5) |
| Operation approach | |
| Laparoscopic | 82 (40.8) |
| Open | 119 (59.2) |
| Tumor size (cm) | 11.4 ± 6.6 |
| Ascites | |
| No | 175 (87.1) |
| Yes | 26 (12.9) |
| Pathology | |
| Benign | 81 (40.3) |
| Borderline | 31 (15.4) |
| Malignancy | 89 (44.3) |
| The number of CTCs | 1.2 ± 2.1 |
| CTC | |
| No | 85 (42.3) |
| Yes | 116 (57.7) |

Values are presented as mean ± standard deviation or number of patients (%).

CA, cancer antigen; CT, computerized tomography; MRI, magnetic resonance imaging; CTC, circulating tumor cells

125 (sensitivity, specificity, accuracy: 78.7%, 53.1%, 66.5%, but lower than CT/MRI (94.2%, 77.9%, 86.5%). When CTC positive or CA-125 > 35 U/mL was defined as "CTC or CA-125" positive, the sensitivity increased to 91.0% and was comparable to that of CT/MRI. When both CTC positive and CA-125 > 35U/mL were defined as "CTC and CA-125" positive, specificity increased to 77.8%, similar to CT/MRI. The AUC (95% confidence interval [CI]; p-value) of CTCs to differentiate ovarian cancer from benign tumors was 0.651 (0.566–0.736; 0.001), which was similar to that of CA-125 (0.656 [0.571–0.741; 0.001]), "CTC and CA-125" (0.691 [0.609–0.773; <0.001]), and "CTC or CA-125" (0.616 [0.529–0.703; 0.011]) and lower than that of CT/MRI (0.861 [0.798–0.923; <0.001]) (Fig 3A). Even when borderline tumors were included (ovarian cancer vs. benign-to-borderline tumors), the sensitivity, specificity, accuracy, and AUC of all modalities were similar to those when borderline tumors were excluded.

For detecting stage I/II ovarian cancer from benign tumors, the sensitivity of CTCs, CA-125, CT/MRI, "CTC and CA-125", and "CTC or CA-125" was 69.6%, 58.7%, 88.6%, 45.7%, and 82.6%, respectively. The specificity of CTCs, CA-125, CT/MRI, "CTC and CA-125", and "CTC or CA-125" was 58.0%, 53.1%, 77.9%, 77.8%, and 33.3%. The AUCs (95% CI; p-value) of CTCs, CA-125, CT/MRI, "CTC and CA-125", and "CTC or CA-125" were 0.620 (0.517–0.723; 0.028), 0.550 (0.444–0.657; 0.358), 0.833 (0.775–0.910; <0.001), 0.599 (0.492–0.706; 0.071), and 0.571 (0.467–0.676; 0.192), respectively (Fig 3B). The sensitivity of all modalities for differentiating stage III/IV ovarian cancer from benign tumors was higher than that for

**Table 2. Characteristics for patients with benign vs. borderline vs. malignant ovarian tumor.**

| | Benign (n = 81) | Borderline (n = 31) | Malignancy (n = 89) | Pᵃ /Pᵇ |
|---|---|---|---|---|
| Age (years) | | | | 0.001/0.133 |
| ≤ 48 | 53 (65.4) | 17 (54.8) | 35 (39.3) | |
| > 48 | 28 (34.6) | 14 (45.2) | 54 (60.7) | |
| CA-125 (U/mL) | | | | <0.001/<0.001 |
| ≤ 35 | 43 (53.1) | 24 (77.4) | 19 (21.3) | |
| >35 | 38 (46.9) | 7 (22.6) | 70 (78.7) | |
| CT or MRI* | | | | <0.001/<0.001 |
| Benign | 50 (64.9) | 5 (16.1) | 1 (1.2) | |
| Borderline | 10 (13.0) | 12 (38.7) | 4 (4.7) | |
| Malignancy | 17 (22.1) | 14 (45.2) | 81 (94.2) | |
| Operation approach | | | | <0.001/0.001 |
| Laparoscopic | 51 (63.0) | 15 (48.4) | 16 (18.0) | |
| Open | 30 (37.0) | 16 (51.6) | 73 (82.0) | |
| Tumor size (cm) | | | | 0.004/0.144 |
| ≤ 11 | 61 (75.3) | 12 (38.7) | 48 (53.9) | |
| > 11 | 20 (24.7) | 19 (61.3) | 41 (46.1) | |
| Ascites | | | | <0.001/0.121 |
| No | 79 (97.5) | 28 (90.3) | 68 (76.4) | |
| Yes | 2 (2.5) | 3 (9.7) | 21 (23.6) | |
| FIGO stage | | | | |
| I | | | 41 (46.1) | |
| II | | | 5 (5.6) | |
| III | | | 27 (30.3) | |
| IV | | | 16 (18.0) | |
| Histology | | | | |
| High grade serous | | | 39 (43.8) | |
| Clear cell | | | 17 (19.1) | |
| Endometrioid | | | 15 (16.9) | |
| Low grade serous | | | 5 (5.6) | |
| Mucinous | | | 6 (6.7) | |
| Other cancer | | | 7 (7.9) | |

Values are presented as mean ± standard deviation or number (%).

* 7 missing

Pᵃ: benign vs. malignant

Pᵇ: borderline vs. malignant

CA, cancer antigen; CT, computerized tomography; MRI, magnetic resonance imaging; FIGO, International Federation of Gynecology and Obstetrics

differentiating stage I/II ovarian cancer. The sensitivities of CTC, CA-125, and CT/MRI for differentiating stage III/IV ovarian cancer from benign tumors were 81.4%, 100.0%, and 100.0%, respectively. The AUCs (95% CI; p-value) of CTC, CA-125, CT/MRI, "CTC and CA-125", and "CTC or CA-125" were 0.684 (0.586–0.782; 0.001), 0.766 (0.684–0.849; <0.001), and 0.890 (0.831–0.949; <0.001), 0.788 (0.700–0.876; <0.001), 0.662 (0.567–0.758; 0.004), respectively (Fig 3C). Regardless of the stage, CT/MRI showed the highest sensitivity and specificity among the single tests. The sensitivity increased with "CTC or CA-125" and the specificity increased with "CTC and CA-125," showing similar values to CT/MRI.

**Table 3. Preoperative CTCs according to final pathologiccal results.**

|  | Benign | Borderline | Cancer | Cancer, early stage | Cancer, advanced stage | P^a/P^b/P^c/P^d |
|---|---|---|---|---|---|---|
| The number of CTCs | 1.09 ± 2.74 | 0.90 ± 1.45 | 1.51 ± 1.48 | 1.30 ± 1.13 | 1.72 ± 1.76 | 0.211/0.052/0.608/0.172 |
| CTC |  |  |  |  |  | <0.001/0.006/0.003/<0.001 |
| No | 47 (58.0) | 16 (51.6) | 22 (24.7) | 14 (30.4) | 8 (18.6) |  |
| Yes | 34 (42.0) | 15 (48.4) | 67 (75.3) | 32 (69.6) | 35 (81.4) |  |

Values are presented as mean ± standard deviation or number (%).

P[a] benign vs. cancer

P[b] borderline vs. cancer

P[c] benign vs. cancer, early stage

P[d] benign vs. cancer, advanced stage

CTC, circulating tumor cells

For differentiating HGS ovarian cancer from benign tumor, the sensitivity of CTCs, CA-125, CT/MRI, "CTC and CA-125", and "CTC or CA-125" was 84.6%, 97.4%, 100.0%, 82.1%, and 100.0%, respectively. The AUCs (95% CI; p-value) of CTCs, CA-125, CT/MRI, "CTC and CA-125", and "CTC or CA-125" were 0.702 (0.605–0.800; <0.001), 0.753 (0.667–0.840; <0.001), 0.890 (0.830–0.949; <0.001), 0.793 (0.704–0.882; <0.001), and 0.662 (0.565–0.759; 0.004), respectively. The sensitivity, accuracy, and AUC of all the modalities for distinguishing non-HGS ovarian cancer from benign tumors were lower than those for distinguishing HGS ovarian cancer (Fig 3D and 3E).

A comparison of the clinical factors according to the presence of preoperative CTCs is presented in S2 Table. CA-125 levels > 35 U/mL (p = 0.013), CT/MRI findings of suspected malignancy (p = 0.004), and moderate-to-severe ascites (p = 0.034) were significantly associated with the presence of CTCs. In the univariate logistic analysis, CA-125 > 35 U/mL (hazard ratio [HR], 2.058; 95% CI, 1.162–3.643; p = 0.013), CT/MRI findings of suspected malignancy (HR, 2.347; 95% CI, 1.306–4.217; p = 0.004), and ascites (HR, 2.743; 95% CI, 1.051–7.162; p = 0.039) were significantly associated with CTC positivity. Multivariate logistic regression analysis showed that CT/MRI finding suspicious for malignancy (HR, 2.347; 95% CI, 1.306–4.217; p = 0.004) was the only independent risk factor associated with the presence of CTCs (S3 Table).

## Discussion

This study showed that the sensitivity, specificity, and accuracy of CTC alone using the TSF platform were comparable to those of CA-125 alone in differentiating ovarian cancer from benign tumors. The diagnostic performance of CTC alone or CA-125 alone was lower than that of CT/MRI. However, this could be overcome by combining CTC and CA-125, regardless of stage or histology. "CTC or CA125" showed increased sensitivity and "CTC and CA-125" showed increased specificity, comparable to CT/MRI. In particular, the sensitivity of "CTC or CA-125" for detecting stage III/IV ovarian cancer or HGSC ovarian cancer was 100.0%, the same as CT/MRI.

A previous study on CTCs as a predictive biomarker in 49 patients with pelvic mass reported that the sensitivity of CTCs for predicting malignancy was 25.7% and the specificity of CTCs for identifying malignancy was 100%. CTC detection rates in non-ovarian origin cancer and primary ovarian cancer were 80% and 17.2%, respectively. There were two differences in the results of our study. First, the study included 6 cases (12%) of non-ovarian origin cancer. Second, the CellSearch method was used for CTC enumeration. The CellSearch system only

**Table 4. Diagnostic performance of preoperative CTC, CA-125, and CT/MRI to evaluate adnexal mass.**

| | Sensitivity | Specificity | Accuracy |
|---|---|---|---|
| Benign vs. cancer (n = 170) | | | |
| CTC | 75.3 (65.0–83.8) | 58.0 (46.5–68.9) | 67.1 (59.5–74.1) |
| CA-125 | 78.7 (68.7–86.6) | 53.1 (41.7–64.3) | 66.5 (58.8–73.5) |
| CT/MRI | 94.2 (87.0–98.1) | 77.9 (67.0–86.6) | 86.5 (80.3–91.3) |
| CTC and CA-125 | 62.9 (52.0–72.9) | 77.8 (67.2–86.3) | 70.0 (62.5–76.8) |
| CTC or CA-125 | 91.0 (83.1–96.0) | 33.3 (23.2–44.7) | 63.5 (55.8–70.7) |
| Benign to borderline vs. cancer (n = 201) | | | |
| CTC | 75.3 (65.0–83.8) | 56.3 (46.6–65.6) | 64.7 (57.6–71.3) |
| CA-125 | 78.7 (68.7–86.6) | 59.8 (50.1–69.0) | 68.2 (61.2–74.5) |
| CT/MRI | 94.2 (87.0–98.1) | 71.3 (61.8–79.6) | 81.4 (75.3–86.7) |
| CTC and CA-125 | 62.9 (52.0–72.9) | 83.0 (74.8–89.5) | 74.1 (67.5–80.0) |
| CTC or CA-125 | 91.0 (83.1–96.0) | 33.0 (24.4–42.6) | 58.7 (51.6–65.6) |
| Benign vs. stage I/II cancer (n = 127) | | | |
| CTC | 69.6 (54.3–82.3) | 58.0 (46.5–68.9) | 62.2 (53.2–70.7) |
| CA-125 | 58.7 (43.2–73.0) | 53.1 (41.7–64.3) | 55.1 (46.0–64.0) |
| CT/MRI | 88.6 (75.4–96.2) | 77.9 (67.0–86.6) | 81.8 (73.8–88.2) |
| CTC and CA-125 | 45.7 (30.9–61.0) | 77.8 (67.2–86.3) | 66.1 (57.2–74.3) |
| CTC or CA-125 | 82.6 (68.6–92.2) | 33.3 (23.2–44.7) | 51.2 (42.2–60.2) |
| Benign vs. stage III/IV cancer (n = 124) | | | |
| CTC | 81.4 (66.6–91.6) | 58.0 (46.5–68.9) | 66.1 (57.1–74.4) |
| CA-125 | 100.0 (91.8–100.0) | 53.1 (41.7–64.3) | 69.4 (60.4–77.3) |
| CT/MRI | 100.0 (91.6–100.0) | 77.9 (67.0–86.6) | 85.7 (78.1–91.5) |
| CTC and CA-125 | 81.4 (66.6–91.6) | 77.8 (67.2–86.3) | 79.0 (70.8–85.8) |
| CTC or CA-125 | 100.0 (91.8–100.0) | 33.3 (23.2–44.7) | 56.5 (47.3–65.3) |
| Benign vs. HGS cancer (n = 120) | | | |
| CTC | 84.6 (69.5–94.1) | 58.0 (46.5–68.9) | 67.7 (57.5–75.0) |
| CA-125 | 97.4 (86.5–99.9) | 53.1 (41.7–64.3) | 67.5 (58.4–75.8) |
| CT/MRI | 100.0 (91.0–100.0) | 77.9 (67.0–86.6) | 85.3 (77.6–91.2) |
| CTC and CA-125 | 82.1 (66.5–92.5) | 77.8 (67.2–86.3) | 79.2 (70.8–86.0) |
| CTC or CA-125 | 100.0 (91.0–100.0) | 33.3 (23.2–44.7) | 55.0 (45.7–64.1) |
| Benign vs. non-HGS cancer (n = 150) | | | |
| CTC | 68.0 (53.3–80.5) | 57.0 (46.7–66.9) | 60.7 (52.4–68.5) |
| CA-125 | 64.0 (49.2–77.1) | 59.0 (48.7–68.7) | 60.7 (52.4–68.5) |
| CT/MRI | 89.4 (76.9–96.5) | 74.0 (64.0–82.4) | 79.0 (71.4–85.4) |
| CTC and CA-125 | 48.0 (33.7–62.6) | 81.0 (71.9–88.2) | 70.0 (62.0–77.2) |
| CTC or CA-125 | 84.0 (70.9–92.8) | 35.0 (25.7–45.2) | 51.3 (43.1–59.6) |

Values are presented as % (95% confidence interval)

CTC, circulating tumor cells; CA, cancer antigen; CT, computerized tomography; MRI, magnetic resonance imaging; HGS, high-grade serous

targets EpCAM and various cytokeratins and has the limitation of not detecting EpCAM-negative cells [21]. In ovarian cancer, no EpCAM overexpression was observed in 24–45% including 8–10% without EpCAM expression [22, 23]. EpCAM is a pan-carcinoma antigen that is overexpressed in many carcinomas. EpCAM expression was also significantly higher in epithelial ovarian cancer tissues than in normal ovarian tissues. Metastatic and recurrent/chemotherapy-resistant ovarian cancers showed significantly higher levels of EpCAM expression than primary ovarian carcinomas [24]. Overexpression of EpCAM in ovarian cancer differs based

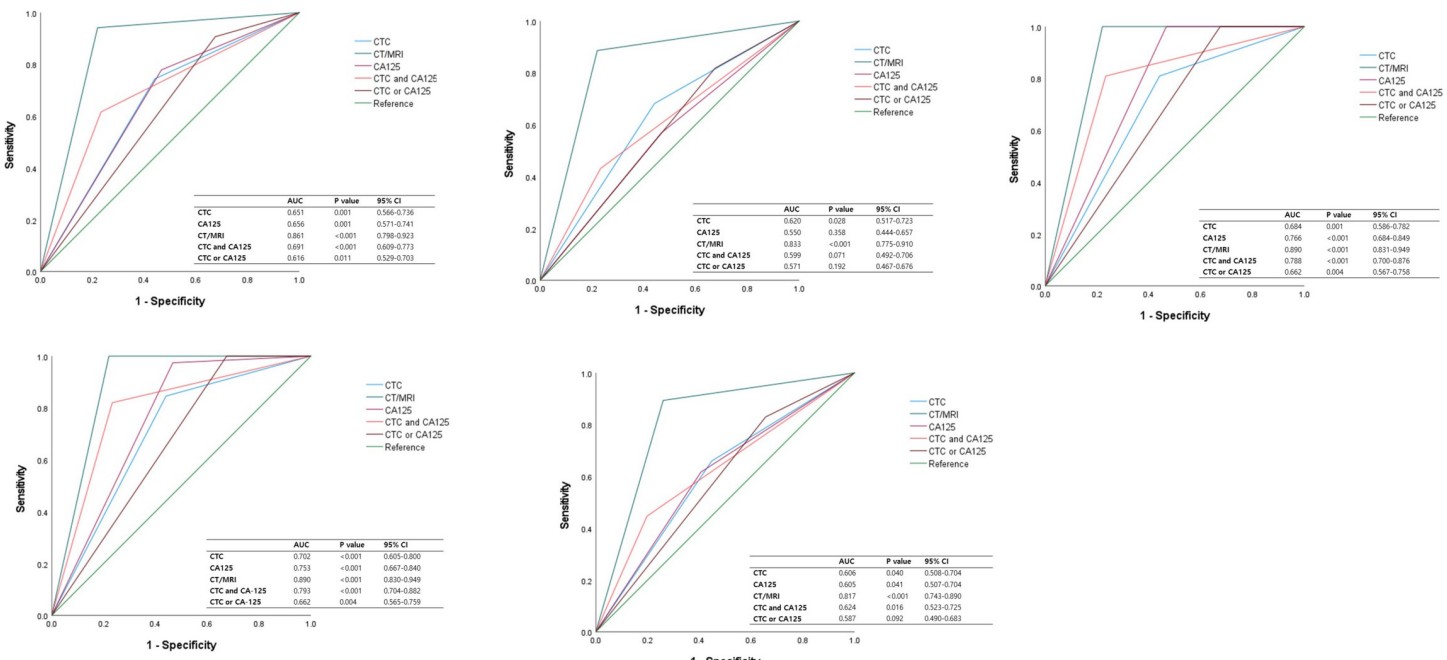

**Fig 3.** Receiver operating characteristic curves of preoperative diagnostic methods in the differential diagnosis of adnexal mass (A) In differentiating ovarian cancer from benign ovarian tumors, excluding borderline ovarian tumors. (B) In differentiating stage I/II ovarian cancer from benign tumor. (C) In differentiating stage III/IV ovarian cancer from benign tumor. (D) In differentiating HGSC ovarian cancer from benign tumor. (E) In differentiating non-HGSC ovarian cancer from benign tumor. CTC, circulating tumor cells; CA, cancer antigen; CT, computerized tomography; MRI, magnetic resonance imaging; AUC, area under the curve; CI, confidence interval.

on the histological type. EpCAM overexpression rate was 55% in mucinous histology and 76% in serous, endometrioid, and other histologies [23]. Another study using the CellSearch system reported a relatively high CTC isolation rate of >80% in patients with epithelial ovarian cancer. Approximately 60% of the study population were patients with recurrent disease, and most (92%) were diagnosed at advanced stage (FIGO stage III or IV) [25]. Application of this CellSearch method to ovarian cancer diagnosis did not seem appropriate because of the varying levels of EpCAM expression in ovarian cancer and the inconsistent results of studies using the CellSearch system.

Several studies have been conducted on CTC detection using various technical approaches in ovarian cancer. In a study using an electrically conductive chip combined with a microfluidic platform, CTCs were detected in 98.1% of ovarian cancer patients (24 with primary and 30 with recurrent diseases). The microfluidic device interacted with antibodies to EpCAM as well as TROP-2, EGFR, vimentin, and N-cadherin [26]. However, the study focused on the predictive effect of CTC detection on ovarian cancer survival rather than on the diagnostic performance of CTCs in ovarian cancer. An alternative approach, the size-dependent isolation of CTCs, was feasible in ovarian cancer and was confirmed by cytomorphological and gene expression analyses [27]. A previous study using tapered-slit membrane filters based on physical properties reported a sensitivity of 77.4% for CTCs in patients with adnexal masses. Particularly, the results suggested that CTCs could be useful for discriminating between early-stage ovarian cancer and benign ovarian tumors with 100% sensitivity [28]. The current study also used the TSF platform for CTC isolation from peripheral blood and reported a similar sensitivity (75.3%) for differentiating ovarian cancer from benign to borderline ovarian tumors. Contrary to the results of a previous study, our study did not demonstrate a good diagnostic performance for early ovarian cancer. Our study showed a higher sensitivity for differentiating

stage III/IV ovarian cancer (81.4%) compared to stage I/II ovarian cancer (69.6%) from benign tumors. The number of cases in our study (89 cases of ovarian cancer and 46 cases of stage I/II ovarian cancer) was greater than that in a previous study (31 cases and 10 cases, respectively). Another study reported that patients with ovarian cancer with a higher FIGO stage had a higher CTC detection rate and that ascites, peritoneal carcinomatosis, and residual disease were significantly associated with high CTC positivity rates [29]. That is, CTCs are considered to reflect the tumor burden of ovarian cancer, and our results also support this.

In animal model studies, CTCs are shown to be more related to hematogenous or lymphatic spread than to the transcoelomic route, which is the main pathway of ovarian cancer spread [30]. Our study showed that CA-125 elevation was not associated with CTC positivity in multivariate logistic regression analysis (S3 Table). CTCs and CA-125 seem to have different pathways that reflect the tumor burden in ovarian cancer. A combination of CTCs and CA-125 can be used complementary for ovarian cancer diagnosis. In our study, this combination also resulted in improving the sensitivity of discriminating between ovarian cancer and benign tumors, as much as CT/MRI.

In our study, the sensitivity of CTCs in the advanced stage and HGSC histologic type was higher than in the early stage and non-HGSC histology. However, CA-125 alone and CT/MRI alone had higher sensitivities than CTC alone for HGSC or advanced-stage ovarian cancer. In advanced-stage ovarian cancer, patients complain of symptoms such as ascites, and CT, MRI, or PET are essential to evaluate the extent of the disease. Therefore, future research on CTCs and their combination with other biomarkers should focus on the differential diagnosis of early-stage and non-HGSC ovarian cancer. Considering that CT, MRI, and PET are expensive, cost-effectiveness analysis is also necessary to apply CTCs, a new non-invasive biomarker, in clinical practice. In addition, based on the increased cell viability obtained using the TSF platform, it can be used for the liquid biopsy of ovarian cancer.

The present study focused on the diagnostic performance of CTC using tapered-slit membrane filter and comparison with other diagnostic method. A strength of this study was the higher number of cases compared to previous studies. The study population had only ovarian-origin masses, excluding benign diseases and cancers of non-ovarian origin, in the final pathologic report. Therefore, our study population was relatively homogeneous compared to those in other studies. However, this study had several limitations. First, blood samples for CTC detection were collected prospectively; however, clinical data were collected retrospectively through a review of medical records. RMI based on ultrasound morphological features and ROMA using two tumor markers (CA-125 and HE4) were not routinely investigated in our hospital. Therefore, CTCs could not be compared with RMI or ROMA, other clinical tools for predicting the malignant potential of adnexal masses. Owing to the subjective interpretation of ultrasound findings, RMI can vary among clinicians, and objective comparison with other methods can be difficult. To objectively compare RMI and CTCs, RMI results evaluated by one experienced clinician are required. Second, the current study did not include analysis of biomarkers other than CTCs and CA-125. To determine the clinical significance of CTCs, it will be necessary to compare the diagnostic performance of other promising biomarkers for ovarian cancer, such as circulating tumor DNA and exosomes, with CTCs. Additional immunohistochemical analyses for tumor-related antigens such as EpCAM or mucin 1 are also needed. Third, there was a time gap in the sample collection peroid for CTCs (between May 2015 and April 2016, and between November 2020 and May 2023) to obtain a larger number of samples. Since we used the same TSF platform and immunofluorescence staining method to isolate CTCs for both time periods, they could be analyzed together.

## Conclusions

While the diagnostic performance of CTC alone using TSF platform in differentiating ovarian cancer from benign tumors was not substantial, the combination of CTC and CA-125 could potentially play a useful role in the diagnostic accuracy of ovarian cancer, similar to that of CT/MRI. Given the substantial costs associated with employing CT/MRI, it is imperative to conduct further comprehensive cost-effectiveness analyses that compare radiologic tests with the combined implementation of CTC and CA-125 for ovarian cancer diagnosis.

## Supporting information

**S1 Table. Characteristics of patients with benign tumor vs. stage I/II ovarian cancer.**
(DOCX)

**S2 Table. Clinical factors according to presence of preoperative CTCs.**
(DOCX)

**S3 Table. Univariate and multivariate logistic regression analyses of risk factors for presence of preoperative CTCs.**
(DOCX)

## Author Contributions

**Conceptualization:** Nam Kyeong Kim, Yong Beom Kim, Young-Ho Cho.

**Data curation:** Nam Kyeong Kim, Minki Kim.

**Formal analysis:** Nam Kyeong Kim, Minki Kim.

**Funding acquisition:** Yong Beom Kim, Young-Ho Cho.

**Investigation:** Nam Kyeong Kim, Minki Kim.

**Methodology:** Yong Beom Kim, Young-Ho Cho.

**Project administration:** Yong Beom Kim.

**Resources:** Dong Hoon Suh, Kidong Kim, Jae Hong No, Yong Beom Kim, Young-Ho Cho.

**Software:** Minki Kim.

**Supervision:** Dong Hoon Suh, Kidong Kim, Jae Hong No, Yong Beom Kim, Young-Ho Cho.

**Validation:** Dong Hoon Suh, Kidong Kim, Jae Hong No, Yong Beom Kim, Young-Ho Cho.

**Visualization:** Nam Kyeong Kim.

**Writing – original draft:** Nam Kyeong Kim.

**Writing – review & editing:** Nam Kyeong Kim, Dong Hoon Suh, Kidong Kim, Jae Hong No, Yong Beom Kim, Minki Kim, Young-Ho Cho.

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
