## [Decision Letter · Decision Letter 0]

10 Apr 2024

PONE-D-23-37967High-throughput viable circulating tumor cell isolation using tapered-slit membrane filter-based chipsets in the differential diagnosis of ovarian tumorsPLOS ONE

Dear Dr. Kim,

Thank you for submitting your manuscript to PLOS ONE. After careful consideration, we feel that it has merit but does not fully meet PLOS ONE’s publication criteria as it currently stands. Therefore, we invite you to submit a revised version of the manuscript that addresses the points raised during the review process.

We look forward to receiving your revised manuscript.

Kind regards,

Girijesh Kumar Patel, PhD

Academic Editor

PLOS ONE

Journal Requirements:

Reviewers' comments:

Reviewer's Responses to Questions

**Comments to the Author**

1. Is the manuscript technically sound, and do the data support the conclusions?

Reviewer #1: Yes

Reviewer #2: Yes

2. Has the statistical analysis been performed appropriately and rigorously? 

Reviewer #1: Yes

Reviewer #2: Yes

3. Have the authors made all data underlying the findings in their manuscript fully available?

Reviewer #1: Yes

Reviewer #2: Yes

4. Is the manuscript presented in an intelligible fashion and written in standard English?

Reviewer #1: Yes

Reviewer #2: Yes

5. Review Comments to the Author

Reviewer #1: Merits

1. 81 (40.3%) benign tumors, 31 (15.4%) borderline tumors, and 89 (44.3%) ovarian cancers were pathologically confirmed.

2. Eighty-one (40.3%), 31 (15.4%), and 89 (44.3%) were pathologically confirmed with benign, borderline, and malignant tumors, respectively.

3. The frequency of age > 48 years (60.7% vs 34.6%, p=0.001) and the frequency of tumor size > 11cm (46.1% vs 24.7%, p=0.004) were significantly higher in the malignant group compared to the benign group.

4. The frequency of abnormal CA-125 was significantly higher in the malignant group compared to the benign and borderline groups.

5. There were significant differences in the frequencies of age > 48 years, tumor size > 171 11 cm, and malignant diagnoses on CT/MRI

Limitations of the study.

1. Blood samples were collected prospectively for the purpose of CTC detection, while clinical data was gathered retrospectively through a thorough review of medical records.

2. some of the biomarkers like circulating tumor DNA ctDNA, Exosomes, EPCAM, MUC1, CEA and HE4 were not analyzed in this study.

3. It should be noted that due to the subjective nature of interpreting ultrasound findings, there may be variations in RMI results among clinicians.

4. Line no 103 - planned (spelling mistake)

Reviewer #2: The present article " High-throughput viable circulating tumor cell isolation using tapered-slit membrane filter-based chipsets in the differential diagnosis of ovarian tumors” by Kim et al provides information regarding isolation of circulating tumor cells (CTCs) using tapered-slit membrane filter (TSF)-based chipsets for the differential diagnosis of adnexal tumor in comparison with other diagnostic method. The study showed that the sensitivity, specificity, and accuracy of CTC alone using the TSF platform were comparable to those of CA-125 alone in differentiating ovarian cancer from benign tumor. The diagnostic performance of CTC alone or CA-125 alone was lower than that of CT/MRI however this could be overcome by combining CTC and CA-125 for improving the diagnostic accuracy of ovarian cancer. Overall, these are the novel observations, and the data support the main conclusion. The overall language in the main text is clear except few minor grammatical errors (Line 103 - Planed surgery). The experimental design was appropriate. The research question was well-defined. Experimental methods were described with sufficient detail. The figures and tables are displayed nicely, and the results supports the conclusion.

6. PLOS authors have the option to publish the peer review history of their article (what does this mean?). If published, this will include your full peer review and any attached files.

Reviewer #1: No

Reviewer #2: No

---

## [Author Response · Author response to Decision Letter 0]

14 Apr 2024

Dear Editor and Reviewers

Ms. No.: PONE-D-23-37967

Title: High-throughput viable circulating tumor cell isolation using tapered-slit membrane filter-based chipsets in the differential diagnosis of ovarian tumors

Dear Sir: 

Thank you very much for the review of our manuscript (Ms. No.: PONE-D-23-37967).

The comments of the review were constructive and have been used to revise and improve the manuscript. We highlighted the edits made to the original version of the manuscript with red color in the revised manuscript. The following is an itemized account of the changes in the manuscript made in response to the comments.

Response to Reviewers

Reviewer #1: 

Reviewer #1: Merits

1. 81 (40.3%) benign tumors, 31 (15.4%) borderline tumors, and 89 (44.3%) ovarian cancers were pathologically confirmed.

2. Eighty-one (40.3%), 31 (15.4%), and 89 (44.3%) were pathologically confirmed with benign, borderline, and malignant tumors, respectively.

3. The frequency of age > 48 years (60.7% vs 34.6%, p=0.001) and the frequency of tumor size > 11cm (46.1% vs 24.7%, p=0.004) were significantly higher in the malignant group compared to the benign group.

4. The frequency of abnormal CA-125 was significantly higher in the malignant group compared to the benign and borderline groups.

5. There were significant differences in the frequencies of age > 48 years, tumor size > 11 cm, and malignant diagnoses on CT/MRI

Author Response: We sincerely thank you for identifying the merits of our study and organizing the key findings.

Limitations of the study.

1. Blood samples were collected prospectively for the purpose of CTC detection, while clinical data was gathered retrospectively through a thorough review of medical records.

Author Response:

We mentioned the above limitation in the Discussion section. In our study, CTCs could not be compared with RMI or ROMA, other clinical tools frequently used to predict the malignant potential of adnexal masses. This was because clinical data were collected retrospectively and preoperative ROMA test or ultrasound results were not prospectively collected in the enrolled patients. To clarify, we have modified this part as follows:

Line 327-330 

RMI based on ultrasound morphological features and ROMA using two tumor markers (CA-125 and HE4) were not routinely investigated in our hospital. Therefore, CTCs could not be compared with RMI or ROMA, other clinical tools for predicting the malignant potential of adnexal masses.

2. some of the biomarkers like circulating tumor DNA ctDNA, Exosomes, EPCAM, MUC1, CEA and HE4 were not analyzed in this study.

Author Response:

As pointed out by reviewer, we have added this limitation to the Discussion section.

Line 333-337

Second, the current study did not include analysis of biomarkers other than CTCs and CA-125. To determine the clinical significance of CTCs, it will be necessary to compare the diagnostic performance of other promising biomarkers for ovarian cancer, such as circulating tumor DNA and exosomes, with CTCs. Additional immunohistochemical analyses for tumor-related antigens such as EpCAM or mucin 1 are also needed.

3. It should be noted that due to the subjective nature of interpreting ultrasound findings, there may be variations in RMI results among clinicians.

Author Response:

We mentioned the above limitation in the Discussion section. We wanted to point out that due to the limitations of RMI testing, caution is needed when comparing RMI and CTCs. To clarify, we have added the following to the Discussion section.

Line 332-333

To objectively compare RMI and CTCs, RMI results evaluated by one experienced clinician are required.

4. Line no 103 - planned (spelling mistake)

Author Response: 

We corrected the spelling error from “planed” to “planned”. (Line 103)

 

Reviewer #2: 

The present article " High-throughput viable circulating tumor cell isolation using tapered-slit membrane filter-based chipsets in the differential diagnosis of ovarian tumors” by Kim et al provides information regarding isolation of circulating tumor cells (CTCs) using tapered-slit membrane filter (TSF)-based chipsets for the differential diagnosis of adnexal tumor in comparison with other diagnostic method. The study showed that the sensitivity, specificity, and accuracy of CTC alone using the TSF platform were comparable to those of CA-125 alone in differentiating ovarian cancer from benign tumor. The diagnostic performance of CTC alone or CA-125 alone was lower than that of CT/MRI however this could be overcome by combining CTC and CA-125 for improving the diagnostic accuracy of ovarian cancer. Overall, these are the novel observations, and the data support the main conclusion. The overall language in the main text is clear except few minor grammatical errors (Line 103 - Planed surgery). The experimental design was appropriate. The research question was well-defined. Experimental methods were described with sufficient detail. The figures and tables are displayed nicely, and the results supports the conclusion.

Author Response: 

We appreciate your review and pointing out the main contents of this manuscript.

We double-checked the entire manuscript for grammatical errors. And then, we corrected the grammatical error from “planed” to “planned”. (Line 103)

---

## [Editor Report · Decision Letter 1]

17 May 2024

High-throughput viable circulating tumor cell isolation using tapered-slit membrane filter-based chipsets in the differential diagnosis of ovarian tumors

PONE-D-23-37967R1

Dear Dr. Kim,

We’re pleased to inform you that your manuscript has been judged scientifically suitable for publication and will be formally accepted for publication once it meets all outstanding technical requirements.

Kind regards,

Girijesh Kumar Patel, PhD

Academic Editor

PLOS ONE
---

## [Editor Report · Acceptance letter]

24 May 2024

PONE-D-23-37967R1 

PLOS ONE

Dear Dr. Kim, 

I'm pleased to inform you that your manuscript has been deemed suitable for publication in PLOS ONE. Congratulations! Your manuscript is now being handed over to our production team.

Kind regards, 

on behalf of

Dr. Girijesh Kumar Patel 

Academic Editor

PLOS ONE